# Efficient Data Management in Agricultural IoT: Compression, Security, and MQTT Protocol Analysis

**DOI:** 10.3390/s24113517

**Published:** 2024-05-30

**Authors:** Mislav Has, Dora Kreković, Mario Kušek, Ivana Podnar Žarko

**Affiliations:** Internet of Things Laboratory, Faculty of Electrical Engineering and Computing, University of Zagreb, HR-10000 Zagreb, Croatia; dora.krekovic@fer.hr (D.K.); mario.kusek@fer.hr (M.K.); ivana.podnar@fer.hr (I.P.Ž.)

**Keywords:** internet of things, precision agriculture, data compression, secure communication, resource optimization

## Abstract

The integration of Internet of Things (IoT) technology into agriculture has revolutionized farming practices by using connected devices and sensors to optimize processes and facilitate sustainable execution. Because most IoT devices have limited resources, the vital requirement to efficiently manage data traffic while ensuring data security in agricultural IoT solutions creates several challenges. Therefore, it is important to study the data amount that IoT protocols generate for resource-constrained devices, as it has a direct impact on the device performance and overall usability of the IoT solution. In this paper, we present a comprehensive study that focuses on optimizing data transmission in agricultural IoT solutions with the use of compression algorithms and secure technologies. Through experimentation and analysis, we evaluate different approaches to minimize data traffic while protecting sensitive agricultural data. Our results highlight the effectiveness of compression algorithms, especially Huffman coding, in reducing data size and optimizing resource usage. In addition, the integration of encryption techniques, such as AES, provides the security of the transmitted data without incurring significant overhead. By assessing different communication scenarios, we identify the most efficient approach, a combination of Huffman encoding and AES encryption, to strike a balance between data security and transmission efficiency.

## 1. Introduction

The Internet of Things (IoT) has changed the way devices interact and exchange data in various areas—from smart homes to industrial automation. At the core of this connected ecosystem are communication protocols that serve as the foundation for seamless connectivity and data exchange between heterogeneous and resource-constrained devices. The convergence of IoT and agriculture has led to a transformative approach known as precision agriculture (PA), which uses connected devices and sensors to revolutionize conventional farming practices [1].

PA represents a cutting-edge methodology in modern agriculture and embodies a data-driven paradigm that integrates a spectrum of ICT solutions, including remote sensing, IoT, and Artificial Intelligence (AI). The main goal is to optimize crop yields and increase agricultural profitability while minimizing the use of resources, including water, fertilizers, herbicides and insecticides [2,3]. This fusion of technologies enables precise data collection and real-time analysis, and the automation of production processes and decision-making to maximize profitability and increase crop productivity while promoting environmental sustainability [4].

The rise in PA has led to an explosion of data in agricultural ecosystems. This is facilitated by seamless data transfer protocols enabled by the IoT. Digital agricultural technologies rely on data collection, storage, integration, and analysis. These technologies use these data to predict events, make recommendations, and even develop automated tools like decision-making robots. Ultimately, they empower farmers with data-driven decision-making. However, a key challenge emerges: many of these technologies are data-intensive, requiring large amounts of data for accurate operation. While these data are crucial, their collection raises privacy concerns for farmers. The sensitivity of agricultural data arises from its diverse and complex nature, which encompasses personal, proprietary, legal and societal interests. Personal and commercial information embedded in agricultural datasets, including the farmer’s identity, financial data, and operational strategies, make them vulnerable to identity theft, fraud, and exploitation. In addition, agricultural data often contain proprietary insights into farming practices, market strategies, and intellectual property, exposing farmers and agricultural businesses to the risk of data misuse and intellectual property theft. In addition, the integrity and traceability of food supply chains rely heavily on the confidentiality and security of agricultural data, as it plays an important role in ensuring food safety standards and consumer confidence. Risks such as data breaches and unauthorized data access pose a significant threat to the confidentiality, integrity and availability of agricultural data and underline the need for security measures. Overall, protecting the confidentiality, integrity, and availability of agricultural data is critical to safeguarding the interests of farmers, consumers, and the entire agricultural ecosystem [5].

Addressing the challenge of managing data traffic in PA, particularly in resource-constrained rural areas with unstable internet connectivity, is crucial for ensuring seamless communication between IoT devices and central cloud systems. The transmission of large data volumes can strain network bandwidth and incur substantial costs, emphasizing the importance of minimizing data traffic to enhance operational efficiency and facilitate timely decision-making [6].

To overcome these challenges, data compression is a viable solution, especially in resource-constrained IoT environments. Compression algorithms offer the ability to reduce the size of data packets transmitted between devices and servers, optimizing resource usage and saving bandwidth. This approach is particularly beneficial in IoT environments, where devices often have limited processing power and storage capacity. By running compression algorithms directly on these resource-constrained IoT devices, the overhead associated with data transmission can be minimized without compromising the integrity or security of the transmitted data. This not only speeds up transmission speeds but also improves the efficiency of data exchange in agricultural networks, ultimately contributing to improved operational performance and cost efficiency [7]. Additionally, encryption techniques play a critical role in safeguarding data transmitted over IoT networks, preserving data integrity and protecting sensitive information from unauthorized access.

In our study, we delve into these challenges and propose strategies to address them effectively. By carefully examining compression algorithms, exploring secure technologies, and analyzing messages transmitted over the Message Queuing Telemetry Transport (MQTT) protocol, we aim to improve the reliability and efficiency of communication and data transmission in PA environments. Leveraging real messages sent from agricultural stations implemented on STM32 microcontrollers, our research bridges the gap between security and efficient data transmission in IoT-based precision agriculture. Our motivation stems from the imperative need to optimize data transmission while ensuring data security, especially in the resource-constrained environments described earlier. Our contributions are as follows:We address the dual challenge of optimizing data transmission and ensuring data security in IoT-based precision agriculture.We conduct experimentation and analysis to evaluate compression algorithms and secure technologies aimed at reducing data traffic while preserving the integrity of sensitive agricultural data.By demonstrating the effectiveness of compression algorithms in reducing data size and integrating encryption techniques to ensure data security, we propose a comprehensive approach to efficient and secure data transmission in agricultural settings.

This article expands upon the initial research findings presented at the 2023 17th International Telecommunications Conference on Telecommunications (ConTEL) [8] with notable improvements, such as a comprehensive analysis spanning various sectors. This includes conducting tests with an increased volume of authentic data transmitted from agrometeorological stations. Additionally, to enhance message size reduction outcomes, message compression algorithms were implemented in this study. Moreover, the analysis of MQTT considered varying levels of delivery guarantees, and with all this in mind, we came up with a new approach to data management in the field of agricultural IoT.

The paper is structured as follows: In Section 2, we provide an overview of related work to put our research into context. Section 3 focuses on compression algorithms and describes their usage. Secure technologies that are essential for protecting IoT devices are discussed in Section 4. Section 5 focuses on the experimental environment, including the specific use case used for the evaluation. Then, Section 6 presents the results of our analysis and the comparison of the different approaches. In Section 7, we provide a discussion on the implications of our results. Finally, in Section 8, we summarize the main findings of our work.

## 2. Related Work

A large body of work investigates methods for reducing the amount of data generated in IoT solutions and solutions for secure communication of end devices with the server. In addition, there are papers that compare different compression algorithms that can be used to reduce content size. Here, we list the papers that are most relevant to our work.

In [9], the authors conducted a comparative study of the performance of the MQTT protocol, comparing the performance variables for a range of payload sizes and security levels. Preliminary results indicate that the use of higher security levels does not result in significant latency overheads when the payload size remains small. Furthermore, they found that the implementation of mutual authentication via Transport Layer Security (TLS) has no impact on MQTT response times for persistent connections when compared to the standard security level where only the server is authenticated. Similar to this paper, in [10], the performance analysis of the MQTT protocol as well as the Constrained Application Protocol (COAP) protocol is presented. This analysis aims to investigate the features and capabilities of the two protocols and determine their feasibility under constrained devices considering security support and various network conditions. Similar investigations and comparisons of different protocols can also be found in other works where various protocols are implemented on an IoT device. In work [11], a comparison was made on an IoT device following the MQTT protocol with different levels of Quality of Service and implementation with and without TLS. Energy consumption was measured, with the results showing that ensuring communication does not cause too much overhead. Similar results were obtained in work [12], where the data transmission time was measured in addition to energy efficiency.

In addition, ref. [13] investigated the real-time properties of IoT communication security protocols in the context of microgrids. It shows the impact of IoT protocols on the real-time requirements of smart grid functions (protection, control, and monitoring) and measures the communication traffic and delay to assess protocol scalability. Three IoT communication protocols were considered in the study: CoAP/DTLS, MQTT/TLS and XMPP/TLS. The results show that both latency and overhead increase by at least three times for each protocol when security mechanisms are applied. The increase in delay was below the recommended maximum standard for the monitoring of the microgrid but exceeded the recommended standard for control operations.

When talking about a security options, another study described in [14] discusses the root causes of MQTT security, the main threats facing MQTT, and the corresponding attack and defense strategies. This includes machine learning-based MQTT security, replay attacks, man-in-the-middle attacks, anomaly detection, the mutual trust mechanism supported by the blockchain in the MQTT network, DoS attacks, and encrypted transmission. So, this article presented the most important MQTT security issues that should be addressed and could be relevant to our approach to MQTT security. Moreover, the authors in [15] proposed a secure version of MQTT and MQTT-SN protocols (SMQTT and SMQTT-SN), where the existing MQTT protocol is extended with a security feature based on Key/Ciphertext Policy-Attribute Based Encryption (KP/CP-ABE) using lightweight elliptic curve cryptography. They also demonstrated the feasibility of SMQTT and SMQTT-SN protocols for various IoT requirements through simulations, evaluating performance in terms of the time taken to perform encryption and the payload size of the packet exchanged in the communication.

Taking a slightly different angle on the topic of performance and security in IoT, some researchers are working on implementing blockchain in IoT systems. Researchers at [16] conducted a comparison with and without blockchain in an IoT system using MQTT as the communication protocol. The test results show that the IoT system with blockchain technology has a higher level of security than the IoT system without blockchain technology. In the overview [17], the researchers also summarized the usage paradigms and open questions regarding the use of blockchain for building reliable IoT systems. This highlights the possibility of using blockchain in some IoT use cases, which also applies to precision agriculture. In this context, ref. [18] presented novel blockchain models that can be used as important solutions to key challenges in IoT-based precision agriculture systems.

In the context of algorithms for compressing smaller content, ref. [19] presented the idea of using a complex of two algorithms, originally used for data compression, as a standard length encryption algorithm to increase the transmission security of very short data sequences, which may even be only one or a few bytes long. This paper presented an approach that uses LZW and Huffman coding to obfuscate data transmission and achieve a basic level of security.

Although our study primarily focuses on other aspects, it is essential to acknowledge the potential of deploying compression algorithms in IoT settings, especially where resource constraints present significant challenges. In the realm of IoT, devices vary widely in computational power, spanning from modest to highly advanced capabilities. Despite the challenges posed by less powerful devices, specialized compression algorithms cater to such scenarios, exemplified by S-LZW [20], tailored for resource-constrained environments. While power and computational limitations present challenges, they do not preclude the implementation of security and compression on IoT devices. In [21], the authors demonstrated compression techniques on an IoT energy meter, facilitating efficient data transmission to the cloud. Additionally, ref. [22] delved into the usability of various compression algorithms on IoT sensor nodes, providing valuable insights into performance optimization and resource consumption. Furthermore, ref. [23] introduced a dual subsystem approach for IoT data compression and transmission, underscoring the effectiveness of compression algorithms in reducing data payload sizes. Lastly, ref. [24] detailed the implementation of hybrid compression techniques on IoT devices, highlighting the pivotal role of efficient algorithms in mitigating resource constraints. Despite the challenges stemming from constrained resources, multiple studies have demonstrated the efficacy of compression techniques in streamlining data transmission and storage within IoT ecosystems. These findings highlight the feasibility and practicality of implementing compression algorithms.

Our work stands out from previous research by focusing on securing MQTT while reducing data for devices in precision agriculture. While previous studies offer valuable insights into the performance, security, and compression of MQTT, our approach is unique in that it focuses on tailoring solutions to the specific needs of precision agriculture. By focusing on this specialized area, we aim to develop innovative strategies that increase MQTT security and improve data transmission efficiency to meet the key requirements of precision agriculture.

## 3. Compression Algorithms

In the interconnected world of the IoT, effective data management is essential, particularly when it comes to reducing the size of the generated data. Data compression is a technique that uses coding rules to significantly reduce the total number of bits needed to store or transfer data. As the amount of information increases, so do the costs of data storage and transmission. With data compression, the information is encoded in fewer bits than in its original form, resulting in less storage space required and shorter transmission times when communicating over a network.

Data compression is popular in sensor networks, as it allows to reduce the volume of data transferred, and thus conserves (a) the battery resource, and (b) the network bandwidth [25]. In IoT sensor nodes, particularly with delta (difference) encoding preprocessing, more frequent occurrences of smaller-valued readings are expected. The compression algorithms employed in this study were chosen for their exceptional performance in compressing streams of such data, especially when the probability distribution of values is unknown in advance. Apart from achieving a favorable compression ratio, these algorithms offer a lossless encoding process that is conducive to embedded devices. They are computationally simple with low space and time complexity, aligning well with the characteristics of IoT devices in general [26].

As the primary focus of the article is not on the algorithms themselves, but rather on the application of data reduction techniques suitable for end devices and the evaluation of traffic reduction, we selected four widely known and utilized algorithms for comparison. We then compared their performance to compress real sensor data typically generated by precision agriculture stations. In the following subsections, we will introduce these algorithms and present a brief overview of how they operate.

### 3.1. Huffman Algorithm

Huffman coding [27], a statistical compression method developed by David A. Huffman, assigns variable-length codes to data symbols based on their frequency. The algorithm efficiently encodes symbols that occur more frequently with shorter bit sequences. This method involves creating a Huffman tree from the input characters and assigning codes by traversing the tree, ensuring unique encoding with distinct bit patterns for each character.

Some of the main advantages of Huffman code are as follows:Variable-Length Encoding: This feature allows for the assignment of shorter codes to more frequently occurring characters, thereby optimizing storage space utilization.Prefix-Free Code: Huffman coding ensures unambiguous decoding by employing a prefix-free code, where no code serves as a prefix of another.Computational Simplicity: The algorithm’s simplicity contributes to fast and memory-efficient encoding and decoding processes.Optimal Average Length: Huffman coding generates code words with the shortest average length, which in turn enhances access time efficiency.

Huffman codes can be fixed (static) or dynamic, used in the deflate method for compression. Fixed codes are based on typical code lengths, while dynamic codes are generated block-wise for input data.

This algorithm is widely utilized in IoT applications ([28,29]), thanks to its computational simplicity and proven effectiveness in achieving efficient compression. Its ability to assign shorter codes to more frequent symbols makes it a reliable choice for reducing data size in various IoT use cases.

### 3.2. Zlib Algorithm

The zlib algorithm [30] is a widely used and versatile compression algorithm that enables lossless data compression. It was developed by Jean-Loup Gailly and Mark Adler and is an integral part of the widely used zlib library. The zlib algorithm, shown in Figure 1, uses the DEFLATE algorithm [31], which combines the LZ77 algorithm for string matching and Huffman coding for symbol coding. DEFLATE achieves compression by replacing repeated occurrences of data with references to a single copy of that data elsewhere in the compressed data stream. This method is particularly effective for compressing text files, executable files, and other types of data with patterns or repetitions, such as sensor readings. One of the notable features of zlib is its high compression ratio, meaning it can significantly reduce the size of files without losing any data. This makes it an excellent choice for scenarios where storage space or bandwidth is limited [32]. In addition, zlib is designed to be efficient in terms of speed and memory utilization.

The implementation of both static (fixed) and dynamic Huffman encoding in zlib adds to its versatility. When choosing between static and dynamic Huffman encoding for each block of LZ77 encoded data, zlib selects the method that produces the smallest output. In cases where both methods yield the same number of bits, zlib defaults to static Huffman encoding for its faster decoding process. This combination of features makes zlib a powerful and efficient tool for data compression needs across various applications [33].

### 3.3. LZW Algorithm

The Lempel–Ziv–Welch (LZW) algorithm [34] is a lossless data compression algorithm developed by Abraham Lempel, Jacob Ziv and Terry Welch. It was first described in 1984 and quickly gained popularity, as it achieves good compression rates. The LZW algorithm works by replacing data sequences with codes, which are then represented with fewer bits.

The key idea behind LZW is to create a dictionary of frequently occurring character sequences during the compression process. As the algorithm processes the input data, it searches for sequences that are already known and adds them to the dictionary. When a new sequence is encountered, it is also added to the dictionary, and the algorithm outputs the code for the longest sequence found thus far. Lengthy recurring sequences are initially stored in the dictionary and become accessible as individual codes for subsequent encoding steps. The described process can be seen in Figure 2. LZW works effectively on data with repetitive patterns, resulting in minimal compression at the start of a message. However, the compression ratio gradually increases as the message size grows, approaching an asymptotic limit [35].

During decompression, the algorithm uses the generated dictionary to reverse the compression process and reconstruct the original data. This method ensures that no information is lost during compression and decompression, making LZW a lossless compression algorithm.

One of the most notable features of LZW is its adaptive nature, i.e., the dictionary is built and updated dynamically as the input data are processed. Thanks to this adaptability, the algorithm can achieve good compression rates for a wide range of input data types and is therefore suitable for different applications [36].

### 3.4. Golomb–Rice Algorithm

Golomb coding, created by Solomon W. Golomb, is a collection of data compression codes tailored for alphabets following a geometric distribution. This type of distribution often occurs in scenarios where smaller values are more common in the dataset. Golomb coding provides an efficient way to represent such data, using variable-length codes. A specific variant of Golomb coding is Rice coding, which was introduced by Robert F. Rice. Rice coding is particularly notable because its parameter is a power of two [37]. This parameterization simplifies the encoding and decoding processes, making it efficient for computer-based applications where binary operations are prevalent.

As mentioned, the Golomb–Rice algorithm utilizes a set of variable-length codes to achieve compression. Variable length coding is a compression method in which shorter code words are assigned to data that are expected to occur with higher frequency. This approach is especially useful for compressing data where smaller values are more likely than larger ones, enabling an efficient implementation using shifts and masks rather than division and modulo operations [38]. The algorithm depends on a parameter *k*, where $m=2^{k}$. To encode a number *x* using Golomb–Rice coding, perform the following:Calculate the quotient *Q* by shifting *x* right by *k* bits.Calculate the remainder R by performing a bitwise AND operation between *x* and (*M* − 1), where $M=2^{k}$.Represent *Q* as a unary value and *R* as a binary value.

The algorithm leverages the fact that dividing by $M=2^{k}$ can be replaced with bit shifting, which is computationally less expensive. This makes Golomb–Rice coding especially useful in scenarios where computational resources are limited, such as in embedded systems or in applications where efficiency is crucial. The choice of *k* affects the trade-off between the compression ratio and encoding/decoding speed [22], with smaller values of *k* leading to higher compression ratios but potentially slower encoding and decoding, and vice versa for larger values of *k*.

## 4. Secure Communication Technologies

In the ever-expanding landscape of networked devices and networks, ensuring robust security measures is essential to protect sensitive data and maintain trust in digital interactions. Therefore, this section looks at two important components of secure communication: Transport Layer Security (TLS) and the Advanced Encryption Standard (AES). In today’s digital age, where cyber threats are widespread and data breaches are a significant risk, the need for secure communication protocols cannot be overstated. However, it is essential to recognize that optimal security often involves navigating complex trade-offs between security, performance, and resource constraints, especially in the area of IoT devices.

While there are more advanced or specialized security options tailored to specific use cases, this research focuses on comparing and evaluating widely used and well-known approaches such as TLS and AES. In this way, we aim to focus on the practical considerations and implications of using these technologies in resource-constrained IoT environments. In this comparative analysis, we acknowledge that we may not be using the most advanced or optimal security solutions. Rather, our goal is to provide insight into the comparative strengths, weaknesses and practical considerations of TLS and AES, taking into account their applicability and performance impact in IoT scenarios. With this research, we aim to contribute to the broader discourse on secure communication protocols in IoT ecosystems, facilitating informed decision-making and robust security implementations.

### 4.1. TLS

Transport Layer Security (TLS) is a cryptographic protocol that ensures secure communication between a client and a server over a network. It works on the transport layer of the OSI model and is intended to guarantee the confidentiality, integrity, and authenticity of the data transmitted between the two parties [39].

TLS achieves confidentiality by encrypting the data exchanged between the client and server, thus preventing unauthorized parties from intercepting and understanding the information. This encryption is accomplished using symmetric encryption algorithms in which both the client and the server use a secret key to encrypt and decrypt the data. Integrity is guaranteed by cryptographic hash functions and message authentication codes (MACs). These mechanisms create digital signatures for each message exchanged between a client and server so that both parties can check that the data have not been tampered with during transmission. Authenticity is established through the use of digital certificates issued by trusted certification authorities (CA). When a TLS connection is established, the server presents the client with its digital certificate, which contains its public key and other identifying information. The client then verifies the authenticity of the certificate by ensuring that it has been issued by a trusted CA and that the server is in possession of the corresponding private key.

With TLS, a handshake process is initiated between the client and the server when a secure connection is established. During the handshake, the two parties negotiate cryptographic parameters, exchange keys, and authenticate each other’s identity. Once the handshake is complete, a secure TLS session is established, enabling secure data transmission between the client and the server.

A more detailed overview of TLS and how it works is presented in [8]. Despite its efficacy in ensuring security, TLS requires a considerable amount of data exchange. Establishing a secure connection demands multiple steps, involving the exchange of digital certificates and encryption keys. Additionally, TLS inflicts computational overhead on both clients and servers during encryption and decryption processes, potentially slowing communication speeds, particularly when microcontrollers are involved as clients.

### 4.2. AES

AES (Advanced Encryption Standard) is an algorithm for secure communication and data protection. As a symmetric encryption algorithm, AES works with data blocks of a fixed length and offers key sizes of 128, 192, or 256 bits. It is established as the standard for encrypting sensitive data in various applications, including network communication, data storage, and digital transactions.

The AES algorithm works via several key processes. With SubBytes, each byte of the input block is replaced by another byte from a fixed table, improving non-linearity. ShiftRows performs cyclic shifts in each row of the state matrix, providing a widespread dispersal of data within the encryption process. MixColumns treats each column of the state matrix as a polynomial, providing further dispersion and dependence on the input. Finally, AddRoundKey XORs the state matrix with the round key and thus ensures that the ciphertext depends on both the plaintext and the encryption key [40].

The implementation of AES on microcontrollers is a challenge due to the limited computing resources. However, optimizations have made it feasible. Some microcontrollers have special hardware modules for cryptographic operations that relieve the CPU of tasks. Even without hardware acceleration, efficient software implementations are possible through optimized algorithms, code, and power management techniques. In IoT devices, AES plays a crucial role in securing communication between devices and backend servers. Despite resource constraints, AES is still a good choice when it comes to ensuring data security while maintaining performance. Thanks to its widespread adoption and efficient implementation strategies, AES is able to protect sensitive data in a wide range of embedded systems and IoT devices [41].

As for TLS, more detailed information on how AES works can be found in the paper [8]. A key feature of the AES algorithm is padding, which is added to the plaintext before encryption. Padding in AES is essential for handling plaintext whose length is not a multiple of the block size, which is 128 bits for AES. The padding ensures that the plaintext can be split into complete blocks before encryption. The padding scheme most commonly used with AES is PKCS#7 (Public Key Cryptography Standard #7) [42], also known as PKCS5 padding.

In PKCS5 padding, bytes are added to the plaintext so that its length is a multiple of the block size. The value of each byte added is the number of bytes added. For example, we have the following:If the plaintext is already a multiple of the block size, an additional block is added, with each byte containing the value 0 × 10 (16 in decimal), as a full padding block.If the plain text is one byte smaller than a multiple of the block size, a byte with the value 0 × 01 is added.If the plaintext is two bytes smaller than a multiple of the block size, two bytes with the value 0 × 02 are added and so on.

During decryption, the padding is removed by examining the last byte of the last block to determine how many bytes were added as padding. Then, this number of bytes is removed from the end of the plaintext. Padding is crucial to ensure that the plaintext can be encrypted and decrypted correctly without losing information. It is an integral part of AES implementations, especially when working with plaintext of different lengths.

## 5. Experimental Environment

In IoT use cases, various communication protocols play a central role in facilitating seamless connectivity and data exchange between devices and centralized systems, especially in precision agriculture. Third generation and fourth generation networks offer high-speed data transmission, wide-coverage, reliability, and mobile connectivity, supporting the real-time monitoring and control of agricultural processes. Fifth generation networks offer ultra-low latency, high bandwidth, and network slicing capabilities that enable near-instant communication, efficient data transmission, and customizable network resources tailored to specific agricultural applications. Wi-Fi technology offers local area connectivity, cost efficiency, and scalability, making it suitable for indoor environments and localized agricultural operations. NB-IoT facilitates the deployment of battery-powered IoT devices in remote agricultural locations with its low-power operation, extended coverage, and cost efficiency. Optimizing the size of the data exchanged is crucial for all the technologies mentioned, including 3G, 4G, 5G, Wi-Fi, and NB-IoT. Efficient data transmission minimizes bandwidth usage, saves energy and improves the responsiveness of IoT applications, thereby improving overall system performance. In our test environment, the specific choice of protocol is not the main concern. Instead, our focus is on developing techniques and methods that prioritize data efficiency and are independent of the underlying communication protocol, ensuring only that they can be applied to devices like STM32 microcontrollers that can be used in precision agriculture use cases. This approach keeps our solutions versatile and applicable across all communication frameworks, ensuring compatibility and effectiveness regardless of the technology used. This flexibility allows agricultural stakeholders to select the most appropriate protocol for their individual needs while benefiting from our optimized data exchange practices that improve the efficiency and performance of precision agriculture applications.

MQTT (Message Queuing Telemetry Transport) stands out as a highly efficient and reliable protocol for communication between IoT devices and servers. Due to its lightweight nature, it is particularly well suited for constrained environments where IoT devices are commonly found and where resources such as bandwidth and battery power are limited [43]. One of the key strengths of MQTT lies in its publish–subscribe message model. This model enables seamless communication between multiple devices and a server without the need for a direct connection between the devices. Devices can publish data on specific topics, while other devices or servers subscribe to these topics to receive the data they are interested in. This decoupling of the sender and receiver promotes the scalability and flexibility of IoT systems. MQTT supports Quality of Service (QoS) levels, which enables varying degrees of message reliability. With QoS levels from 0 to 2, MQTT can cater to different application requirements and ensure that messages are delivered reliably even under unreliable network conditions.

Additionally, the minimal overhead and low bandwidth utilization of MQTT make it ideal for IoT deployments where resource conservation is critical. Its efficient binary encoding format further reduces the size of transmitted data, minimizing network traffic and saving energy, which is particularly beneficial for battery-powered devices [44]. Furthermore, MQTT supports persistent sessions and retained messages, which ensures that devices can stay connected and receive important information even if they are temporarily offline. This feature is invaluable in IoT scenarios where intermittent connectivity is common.

Overall, MQTT is a preferred choice for communication in IoT ecosystems due to its simplicity, efficiency, and robustness. It enables seamless data exchange between devices and servers while optimizing resource utilization and reliability.

The measurements were performed depending on different comparison criteria:Comparison depending on the Quality of Service: For a simple MQTT plaintext scenario, we performed a comparison depending on the levels of the MQTT parameter Quality of Service, which represents different guarantees of message delivery.Comparison after using the data compression algorithm: For the service level that generates the fewest messages and the least amount of data (QoS = 0), we made a comparison of the different compression algorithms used for compression of the payload.Comparison depending on different security levels: In the last measurement on the implicit level of Quality of Service (QoS = 0) and a sufficient compression using the Huffman algorithm, we used different security approaches. The first approach refers to MQTT-plaintext, without additional security. Another approach involves MQTT with the implemented TLS protocol (MQTTS). The third approach involves the use of MQTT in conjunction with the AES cryptographic protocol.

In order to conduct tests with real data, we collected messages originating from agricultural stations and transmitted them to the server for analysis. From this dataset, we selected a representative subset of 100 messages for compression experimentation. These compression experiments involved the application of algorithms previously described in Section 3. The implementation of these compression algorithms was performed in Java. The obtained results will be presented in the following sections. For communication testing purposes, the Paho client in the Java programming language was used, with MQTT version 3.1.1. The same message was used for all cases, which represents the message that is sent in the smart agriculture use case. The length of this message is 204 bytes. In addition to the same payload, the same MQTT parameters such as the client ID and the topic to which the messages were sent also applied to all cases. An MQTT broker was implemented using the open-source Eclipse Mosquitto. When using TLS, the certification authority (CA) that the client can use to access the broker must be specified in the broker configuration, as well as the server key and the server certificate. The specified files are needed to achieve a minimum level of security through TLS. OpenSSL was also used to generate a CA key pair, a CA certificate, a server key pair, and a server certificate. For data encryption, the AES-128 ECB algorithm was used with an arbitrary secret key of length 16 bytes. Wireshark was used to record the data traffic and analyze the size of the transmitted data. A measurement scenario includes opening a client connection with a broker, publishing on a previously defined topic, and closing the connection.

## 6. Results

As the testing was performed in three different scenarios, the results are also presented in three different subsections, each depending on a different aspect.

### 6.1. Quality of Service

The first test scenario presents a comparative analysis of MQTT message sending depending on different QoS values. Since the QoS itself represents the level of message delivery guarantee, it is clear that a higher delivery guarantee generates a larger number of messages. The delivery guarantee level and specific messages for each QoS level are listed below.

QoS 0 (At most once delivery):–Messages are delivered at most once, meaning that the sender does not expect any acknowledgment or confirmation from the receiver.–This level of QoS provides the lowest overhead and the highest message throughput but offers no guarantee of delivery.QoS 1 (At least once delivery):–This level of QoS ensures that messages are delivered at least once to the receiver.–The sender sends the message and expects to receive an acknowledgment (ACK) from the receiver. If no ACK is received within a certain time frame, the message is re-sent.–QoS 1 guarantees that each message will be delivered at least once, but it may result in duplicate messages being received by the receiver if the ACK is lost or delayed.QoS 2 (Exactly once delivery):–This is the highest level of QoS in MQTT, providing the strongest guarantee of message delivery.–Messages are delivered exactly once to the receiver.–This level of QoS involves a four-step handshake process:*The sender sends the message to the receiver and waits for an acknowledgment (PUBREC).*Upon receiving the message, the receiver sends an acknowledgment (PUBREC) to the sender.*The sender sends a message acknowledgment (PUBREL) to the receiver, indicating that it has received the PUBREC.*Finally, the receiver obtains the acknowledgment (PUBCOMP), completing the process.–QoS 2 ensures no message loss or duplication, but it also involves the highest overhead in terms of message transmission.

The exchanged MQTT messages and their sizes are shown in Table 1.

Looking at the data in the table, it can be seen which messages appear in each of the cases described and how large the messages are. It should be noted that for a complete analysis, we included messages on the IP layer (including TCP handshake and IP packet headers) in the calculation. Therefore, we see in the table that a higher delivery guarantee generates a higher number of messages, which generates a higher number of messages on the TCP layer, and therefore a higher total amount of data traffic.

### 6.2. Compression Algorithm

To reduce the size of messages, in particular, the size of the payload transmitted in the message, the use of a compression algorithm is prescribed. We wanted to find a suitable algorithm for the messages sent when using agricultural stations. Since the success of the compression and the size of the message after compression depend on the content of the message itself, the final result after compression may vary. In our tests, we used 100 messages with partially different content in a key value format, where the key values were the same for each message, while the value changed to match the sensor data of a device in the field. For the reasons mentioned above, we gave the mean value of the size of the content after compression as the result. The results can be seen in Table 2. We define the compression ratio as $K=S_{0}/S_{c}$, where $S_{0}$ is the uncompressed size in bytes and $S_{c}$ is compressed size in bytes, i.e., higher *K* means better compression. Another comparable metric is the reduction percentage, which serves as a measure of the efficiency of a compression algorithm in reducing the data size. This metric is determined by calculating the difference between the original size of the data and the compressed size, dividing it by the original size, and then multiplying it by 100 to express the result as a percentage. This formula, which is widely used in the field of data compression, provides a quantifiable measure of the extent to which the data size has been reduced.

Table 2 compares the average size of data after compression, compression ratios, and reduction percentages for each algorithm. The Huffman algorithm achieved the highest reduction percentage of 32.9%, followed by Zlib with 20.4%. LZW and Golomb–Rice had lower reduction percentages of 4.37% and 3.01%, respectively. The compression of messages can therefore take place before any of the methods mentioned for communication and achieving security. This ensures that the payload, i.e., the MQTT Publish message, is reduced in any of the mentioned cases. In IoT use cases, the Huffman algorithm can outperform Zlib, LZW, and Golomb–Rice due to its efficiency in compressing data with varying symbol frequencies. In scenarios where data contain repetitive patterns or symbols with varying probabilities, Huffman coding excels by assigning shorter codes to more frequent symbols, leading to better compression ratios. Furthermore, Huffman coding is known for its simplicity and speed in both encoding and decoding processes.

### 6.3. Secure Communication Technology

In the third test scenario, we added a level of security to MQTT and performed the same measurements as in the first scenario. In this case, the lowest level of QoS was implied, which was sufficient for use cases and generated the least amount of traffic. In addition, the content sent was compressed before a security technology was used. The assumed value for the size of the user data was reduced from 204 bytes to 137 bytes before the subsequent security mechanisms were used. Therefore, the Huffman algorithm was used for compression, which provided the best results as shown in the previous test case. In the first case, we sent a compressed message without additional security. In the second case, TLS was used as an additional level of security. In the third case, the payload was encrypted using the AES algorithm, and the encrypted content was sent as in the first case.

#### 6.3.1. MQTT-Plaintext

In the first case, there was no additional security, so the message was sent as plaintext. The amount of data generated in this case corresponded to the first case of the first measurement, except that the message was compressed. The size of the Publish message was therefore 196, while the total amount of data traffic on the IP layer was 896 bytes. In this case, fewer data were generated, but there was no security implementation.

#### 6.3.2. MQTT-TLS

In the second test case, the implementation of the TLS protocol introduced additional security measures during the connection establishment and data transmission. Although this increased the security of the communication, it also led to an increase in the number and size of the messages exchanged. In contrast to the standard MQTT handshake shown in the first test case, the TLS handshake took place after a TCP connection had been established. This TLS handshake includes the exchange of the messages shown in Table 3, the definition of the security parameters for further communication between client and server, and the guarantee of confidentiality of the encrypted message content. While the additional security measures offered by MQTT with TLS provide critical protection for sensitive agricultural data, the increased message overhead and volume should be carefully weighed against the security benefits, especially in resource-constrained IoT environments with STM32 devices as is common in precision agriculture.

Client Hello (353 bytes): This message is initiated by the client to establish a connection with the server and initiate the TLS handshake. It contains information about supported cryptographic algorithms, TLS version, and other parameters.Server Hello (1429 bytes): Upon receiving the Client Hello, the server responds with the Server Hello message. This message includes the chosen cryptographic algorithms, TLS version, and other parameters selected by the server for the connection.Change Cipher Spec (124 bytes): This message is sent by both the client and the server to signal the switch to encrypted communication. It confirms that subsequent data will be encrypted using the negotiated parameters.TCP MESSAGES EXCHANGED (728 bytes): This includes any TCP-level messages exchanged during the connection setup and data transmission process. These messages are essential for establishing and maintaining the TCP connection between the client and server.Application Data (1164 bytes): This message contains the actual data being transmitted, such as sensor readings or control commands, within the MQTT payload.TOTAL (3789 bytes): This is the sum of all the message sizes mentioned above. It represents the total size of data exchanged during the process of opening a connection and sending a message in MQTT with TLS.

The table illustrates the breakdown of message sizes involved in the MQTT with TLS communication process. It highlights the overhead introduced by the TLS handshake, TCP messages, and the actual application data, which is crucial to consider for optimizing network performance and resource utilization in IoT applications, including those in precision agriculture.

#### 6.3.3. MQTT-AES

In the third test case, the data transfer was similar to the first case. Instead of implementing the TLS protocol to establish the connection and send messages, the client encrypted the message using the AES algorithm. This encryption protected the content of the message during transmission and enabled decryption by the server or the subscriber using a predefined private key. However, as the AES algorithm dictates that the input must be a multiple of 16 bytes, with padding added if required, the size of the published message can vary.

In our case, where the size of the payload is 204 bytes, it will be 208 bytes after encryption. Compared to the first scenario, only the size of the MQTT Publish message increases by 4 bytes. Consequently, the total size of messages exchanged is 940 bytes, which is 4 bytes larger than in the case of MQTT plaintext. This ensures security during message transmission with minimal overhead.

The following graph in Figure 3 presents a comparative analysis of data volumes in MQTT communication scenarios utilizing different security approaches: MQTT-Plaintext, MQTT-TLS, and MQTT-AES. Each scenario represents a distinct approach to securing data transmission, with MQTT-TLS encrypting all exchanged data, MQTT-AES encrypting message payloads, and MQTT-Plaintext transmitting data without encryption.

The analysis of the data generated in the three test cases shows a clear difference in overhead between the TLS and AES encryption protocols. It turns out that MQTT-TLS generates the highest data volume with 3789 bytes and thus clearly underperforms the two cases MQTT-Plaintext and MQTT-AES. Although MQTT-AES introduces a minimal overhead, resulting in a data volume of 903 bytes, it remains well below that of MQTT-TLS. In contrast, MQTT-Plaintext has the lowest data volume at 896 bytes, which shows that there is no encryption overhead. These results highlight the trade-offs between security and efficiency, with MQTT-TLS offering higher security at the cost of a larger data volume compared to MQTT-AES. Therefore, when selecting a security option, the balance between the security requirements and the impact on the efficiency of data transmission should be evaluated.

## 7. Discussion

According to the results presented, the use of different communication technologies and different options within these technologies leads to a different type of message transmission and thus to a different volume of data traffic generated. Which communication protocol, delivery guarantee, or security technology to choose depends primarily on the use case for which these aspects will be used. In some cases, it will be necessary to ensure the highest level of security regardless of the amount of traffic exchanged, while in other cases, it will be crucial to use as few data as possible with a low level of security and delivery guarantee.

Regardless, our work proposes a combination that satisfies both aspects: an adequate level of communication security and a reduction in the amount of traffic generated during communication. Therefore, the most favorable solution is the one in which the payload containing the information is compressed to reduce the size of the payload by a certain percentage. It is then possible to protect the encrypted payload with a cryptographic algorithm with minimal overhead. Due to the peculiarity of the AES algorithm, which requires the content of the payload to be a multiple of 16 and, conversely, padding to be added, it makes more sense to use compression first and then encryption. It should also be noted that the content of the message received from the server or another client must first be decrypted and then decompressed in order to obtain the original content. The process described with the sequence of actions and the size of the proposed message is shown in Figure 4.

A message that was originally 204 bytes in size was therefore first compressed using one of the compression algorithms. In our case, the best results were achieved with the Huffman algorithm, which can reduce the size of the content to 137 bytes. To ensure security during transmission, the message was then encrypted using the AES algorithm for content encryption. This increased the size of the message content to 144 bytes. The size of the sent MQTT Publish message was 203 bytes. As already mentioned, the receiver of the message must perform the operations in reverse order, i.e., first decrypt and then decompress in order to access the original content. This completed the process, and the receiver obtained the original message of 204 bytes. Scaled to the total size of the exchanged messages, this case generated 876 bytes in total, which is visibly less than the cases presented in the previous section.

In terms of the scalability of the proposed solutions in larger agricultural IoT networks, this solution behaves in the same way, as the introduction of new devices and clients into the system does not entail any changes. As the focus is on achieving security and minimizing data during a single connection, scalability is not an issue in this regard. In this sense, it does not matter how many devices are used, as the same results will apply to each device. On the other hand, for a comprehensive assessment, it is essential to consider the volume of data transmitted over longer periods of time, especially in the context of precision agriculture, where devices regularly connect to servers and transmit messages several times a day. Table 4 shows how the volume of data varies when a single message transmission is extrapolated over daily, monthly, and annual periods. In our analysis, we assumed a scenario where a client sends a message every 10 min. To assess data transmission over longer periods, we selected the most effective approach from the scenarios tested, prioritizing either minimal traffic generation or achieving an acceptable level of security.

So, the scenarios were compared:MQTT-Plaintext—neither additional security nor content compression is used.MQTT-Huffman—Huffman algorithm is used for compression.MQTT-Huffman-AES—compression is used first and then AES encryption.

The results of the data set scaled over a longer period of time show certain differences in the total. Although the figures look very similar at first glance, they differ to a certain extent. Scaled to the annual level, the difference between the first and second cases is 3.43 MB. This is the difference caused by the compression to reduce the amount of data. Adding the security option on a yearly basis introduces an overhead of 0.32 MB, which is an acceptable size for guaranteed security. In cases where we have multiple devices sending data, this number increases significantly. In other words, the use of compression can lead to significant savings in the size of the messages exchanged, which in turn can lead to other savings relevant to the area in which this is used. These results illustrate the trade-offs between data security, transmission efficiency, and resource utilization in MQTT-based communication systems. While plaintext transmission offers simplicity and minimal overhead, it exposes data to potential security risks. AES encryption increases security but results in a slight increase in data size. The combination of Huffman encoding and AES encryption proves to be the most efficient approach, significantly reducing data volume while ensuring data confidentiality.

In addition to the amount of data and the level of security that needs to be implemented, the required processing power of the device itself for each of these solutions could also be considered a factor that may influence the analysis. The required processing power on STM32 devices depends on several factors, including the STM32 variant used, clock frequency, and data sizes, as well as the complexity of the cryptographic operations, encryption and compression algorithms, and key sizes. Regardless, the computing power required for compression algorithms such as Huffman or the computing power for MQTT as plaintext, with TLS or with AES on STM32 devices is generally manageable, but it is also one of the important characteristics to consider.

## 8. Conclusions

In summary, the integration of IoT into agriculture, known as precision agriculture, has revolutionized farming practices by using connected devices and sensors to optimize processes and improve sustainability. The efficient management of data traffic is critical in agricultural environments, given challenges such as limited internet connectivity and bandwidth constraints. Our research focuses on optimizing data transmission in agricultural IoT through compression algorithms and secure technologies to effectively address these challenges. Through experimentation and analysis, we evaluated different approaches to minimize data traffic while ensuring data security. The results show the effectiveness of compression algorithms, especially Huffman coding, in reducing data size and optimizing resource usage. The integration of encryption techniques such as AES ensures data security without significant overhead. When evaluating different communication scenarios, we found that the combination of Huffman coding and AES encryption provides the most efficient approach that balances data security and transmission efficiency. This approach significantly reduces data volume while protecting sensitive agricultural data. Looking to the future, our findings highlight the importance of considering both security requirements and transmission efficiency in IoT agriculture. This enables farmers to improve operational efficiency, mitigate risk, and promote sustainable farming practices in an increasingly connected ecosystem, contributing to the evolution and resilience of agriculture in the digital age.

## Figures and Tables

**Figure 1 sensors-24-03517-f001:**
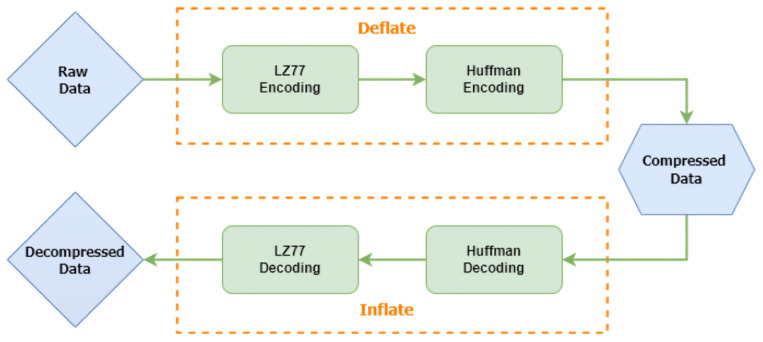
Overview of the zlib encryption and decryption process.

**Figure 2 sensors-24-03517-f002:**
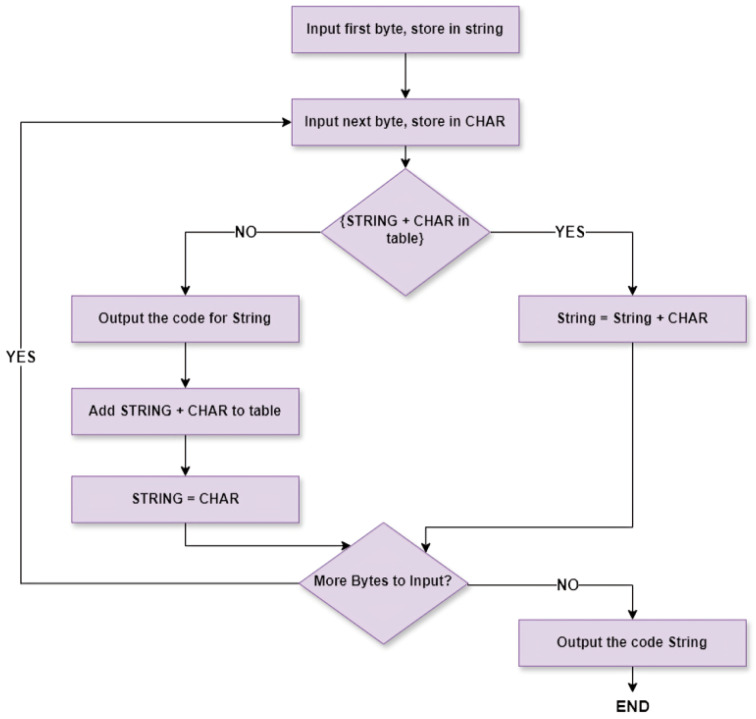
Flowchart of conventional LZW.

**Figure 3 sensors-24-03517-f003:**
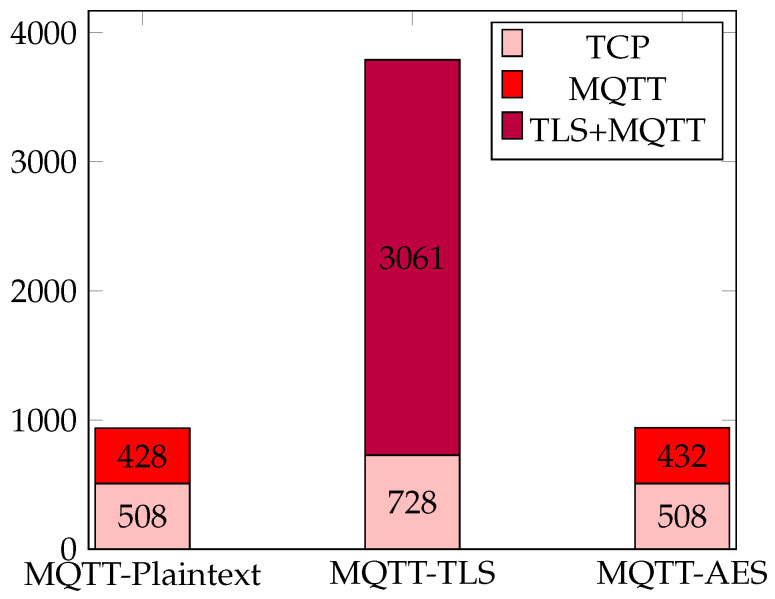
Comparison of data amount in bytes.

**Figure 4 sensors-24-03517-f004:**
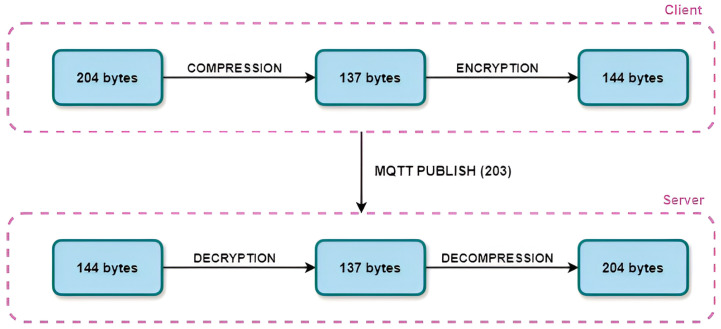
Proposed flow for reduced and secure MQTT message.

**Table 1 sensors-24-03517-t001:** Comparison of MQTT QoS message sizes in bytes.

Message	QoS = 0	QoS = 1	QoS = 2
CONNECT	71	71	71
CONNACK	48	48	48
PUBLISH	263	265	265
PUBACK	-	48	-
PUBREC	-	-	48
PUBREL	-	-	48
PUBCOMP	-	-	48
DISCONNECT	46	46	46
TCP MESSAGES EXCHANGED	508	552	640
TOTAL	936	1030	1214

**Table 2 sensors-24-03517-t002:** Comparison of compression algorithms.

Compression Algorithm	Average Size after Compression	Compression Ratio	Reduction (%)
Huffman	136.89	1.490	32.9
Zlib	162.31	1.257	20.4
LZW	195.09	1.046	4.37
Golomb–Rice	197.85	1.031	3.01

**Table 3 sensors-24-03517-t003:** MQTT-TLS message sizes in bytes.

TLS Message	Size
Client Hello	353
Server Hello	1429
Change Cipher Spec	124
TCP MESSAGES EXCHANGED	728
Application Data	1164
TOTAL	3789

**Table 4 sensors-24-03517-t004:** Comparison of data traffic in MB during longer period.

	1 Day	1 Month	6 Months	1 Year
MQTT-Plaintext	0.14	4.16	24.96	49.92
MQTT-Huffman	0.13	3.87	23.22	46.49
MQTT-Huffman-AES	0.13	3.90	23.41	46.81

## Data Availability

The data that support the findings of this study are available by request from the corresponding author.

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
