# Peer review of "Efficient Data Management in Agricultural IoT: Compression, Security, and MQTT Protocol Analysis"

_sensors, 2024, doi:10.3390/s24113517_

Round 1
Reviewer 1 Report
Comments and Suggestions for Authors
The authors are addressing an important topic. However, the paper lacks scientific merit and primarily serves as a benchmarking report.
There are already several articles in the literature discussing performance improvement techniques and security for MQTT. Perhaps a survey on the state-of-the-art can benefit the authors to have a more specific research problem and contribution.
One of the main reasons that security and compression are not implemented on IoT devices is the power constraint and computation power limitations, which are not addressed in this paper.
The connection and justification using a precision agriculture scenario, benefiting from compression and security, are significantly missed in the paper.
Finally, the authors should elaborate on how agricultural data is sensitive data.
Comments on the Quality of English LanguageIn terms of English, the paper is acceptable. However, the content is very general and not specific to the problem at hand. The presentation could benefit from improving the resolution of the figures.
Reviewer 2 Report
Comments and Suggestions for Authors
This paper presents a comprehensive study on optimizing data transmission in agricultural Internet of Things (IoT) solutions by leveraging compression algorithms, secure technologies, and MQTT protocol analysis. The research focuses on minimizing data traffic while ensuring data security, addressing the challenges faced in agricultural environments. Through experimentation and analysis, the effectiveness of compression algorithms, particularly Huffman coding, in reducing data size and enhancing resource usage is demonstrated. Integration of encryption techniques, such as AES, is also explored to secure transmitted data without significant overhead. The study aims to provide insights for efficient and secure data transmission in precision agriculture.
The paper effectively communicates the research objectives, methodologies, and findings in a clear and structured manner. The research methodology and results are presented accurately, supported by empirical evidence and analysis.
The paper builds upon existing research on data compression, security challenges in MQTT, and IoT protocols for agricultural applications. It contributes by offering a specialized focus on precision agriculture and innovative strategies to enhance data transmission efficiency and security.
The authors would enhance their paper by including a thorough discussion on the potential benefits of integrating blockchain technology into MQTT network security. For example, they should reference:
AI-enhanced blockchain technology: A review of advancements and opportunities
Ressi, D., Romanello, R., Piazza, C., Rossi, S.
Journal of Network and Computer Applications, 2024, 225, 103858
F. Chen, Z. Xiao, L. Cui, Q. Lin, J. Li, S. Yu, Blockchain for internet of things applications: A review and open issues,
Journal of Network and Computer Applications 172 (2020) 102839.
Moreover, I suggest the authors to explore the scalability of the proposed solutions in larger agricultural IoT networks, investigating the impact of real-time data processing on data management, and further enhancing the integration of encryption techniques for enhanced data security.
Finally, additional insights on the practical implications of the research could enhance the paper's impact.
Reviewer 3 Report
Comments and Suggestions for Authors
This paper presents a comprehensive study that focuses on optimizing data transmission in agricultural IoT solutions with the use of compression algorithms and secure technologies. Through experimentation and analysis, we evaluate different approaches to minimize data traffic while protecting sensitive agricultural data. Our results highlight the effectiveness of compression algorithms, especially Huffman coding, in reducing data size and optimizing resource usage. In addition, the integration of encryption techniques, such as AES, provides the security of the transmitted data without incurring significant overhead. By assessing different communication scenarios, we identify the most efficient approach, a combination of Huffman encoding and AES encryption, to strike a balance between data security and transmission efficiency.
The topic is interesting, the novelty and organization of this study are very weak. Extensive revision is required. Detailed comments are given as:
1. The motivation in abstract and introduction should be improved. In addition, research gap should be clearly mentioned in abstract and introduction.
2. It is suggested to provide suitable references in Paragraph 2 Introduction Section.
3. What are the major contributions of this study? What is the importance of this study?
4. It is suggested to combine Section 5 and 6 into one section, called Experimental Results and Analysis.
5. The authors have provided just already existing compression algorithms (several), there is no novelty? Being a comprehensive study, authors should add a section about lesson learned from this study.
6. It is suggested to provide high quality HD figures, as some figures are not of good quality, e.g., Figure 1, 2, etc.
7. Please check the whole manuscript for typos and grammar errors, and improve the language of the paper.
Comments on the Quality of English LanguageModerate editing of English language required
Round 2
Reviewer 1 Report
Comments and Suggestions for Authors
The authors should dedicate more time to familiarize themselves with the security and privacy challenges in Precision Agriculture (PA). Just presenting a generic paragraph doesn't offer much assistance.
Regarding this "The sensitivity of agricultural data arises from its diverse and complex nature, which encompasses personal, proprietary, legal, and societal interests," do the authors truly accept this statement based on the details provided in the paper? The authors mentioned the algorithms they developed for compression, decompression, encryption, and decryption, all of which are utilized in several well-established research studies in the literature.
I highly recommend that the authors consider conducting a survey before making claims, especially those mentioned in the response note. For instance, the statement "even machine learning can be performed on IoT devices, showcasing the potential for compression algorithms to be implemented effectively,"; Does the ability to perform machine learning justify the assumption that compression algorithms can be implemented effectively?
Dear authors, I recommend you to consider a literature review on the following areas:
1. Resource-constrained devices (particulary IoT)
2. Combination of compression and encryption for MQTT
3. Obfuscation algorithms and data transport using MQTT
4. Security and privacy concerns surrounding agricultural data
Comments on the Quality of English Language
The English is all right.
Reviewer 2 Report
Comments and Suggestions for Authors
In my view, the final version of the manuscript should include a discussion on the scalability of the proposed solutions within larger agricultural IoT networks, as well as incorporate the recommended references.
I would like to reconsider the paper before confirming its acceptance.
Reviewer 3 Report
Comments and Suggestions for Authors
Authors have addressed all my concerns in the revised manuscript, It can be accepted for publication.
Author Response
Manuscript accepted for publication.